# Multigranularity Syntax Guidance with Graph Structure for Machine Reading Comprehension

**Chuanyun Xu [1,2,\*], Zixu Liu [1,\*], Gang Li [1], Changpeng Zhu [1] and Yang Zhang [2]**

1 School of Artificial Intelligence, Chongqing University of Technology, Chongqing 400054, China
2 College of Computer and Information Science, Chongqing Normal University, Chongqing 401331, China
* Correspondence: xcy@cqnu.edu.cn (C.X.); lzx5623@2020.cqut.edu.cn (Z.L.)

**Abstract:** In recent years, pre-trained language models, represented by the bidirectional encoder representations from transformers (BERT) model, have achieved remarkable success in machine reading comprehension (MRC). However, limited by the structure of BERT-based MRC models (for example, restrictions on word count), such models cannot effectively integrate significant features, such as syntax relations, semantic connections, and long-distance semantics between sentences, leading to the inability of the available models to better understand the intrinsic connections between text and questions to be answered based on it. In this paper, a multi-granularity syntax guidance (MgSG) module that consists of a "graph with dependence" module and a "graph with entity" module is proposed. MgSG selects both sentence and word granularities to guide the text model to decipher the text. In particular, syntactic constraints are used to guide the text model while exploiting the global nature of graph neural networks to enhance the model's ability to construct long-range semantics. Simultaneously, named entities play an important role in text and answers and focusing on entities can improve the model's understanding of the text's major idea. Ultimately, fusing multiple embedding representations to form a representation yields the semantics of the context and the questions. Experiments demonstrate that the performance of the proposed method on the Stanford Question Answering Dataset is better when compared with the traditional BERT baseline model. The experimental results illustrate that our proposed "MgSG" module effectively utilizes the graph structure to learn the internal features of sentences, solve the problem of long-distance semantics, while effectively improving the performance of PrLM in machine reading comprehension.

**Keywords:** machine reading comprehension; graph attention network; BERT; SQuAD

## 1. Introduction

Machine reading comprehension (MRC) has long been a vital task in natural language processing (NLP). It refers to the ability of machines to read and understand a given text and answer questions based on it (as shown in Figure 1). Better analysis and understanding of the meaning of a sentence has always been a challenge for researchers in natural language understanding (NLU). It is relevant in several ways, for example, in intelligent customer service and speech assistants [1]. MRC has flourished, driven by the growing interest of researchers, as well as the public release of numerous datasets (for example, the Stanford Question Answering Dataset (SQuAD) [2,3]). Current MRC models are expected to perform the tough tasks of not only finding relevant answers to questions based on a certain text but also determining the questions that cannot be answered. For such models to be more adaptable to the real world, a deeper understanding of the text in conjunction with semantics and other information is required.

**Passage**：The English name "Normans" comes from the French words Normans/ Normanz, plural of Normant, modern French normand, which is itself borrowed from Old Low Franconian Nortmann "Northman" or directly from Old Norse Norðmaðr, Latinized variously as Nortmannus, Normannus, or Nordmannus (recorded in Medieval Latin, 9th century) to mean "Norseman, Viking".

**Q1:** What is the original meaning of the word Norman?

**A1:** Viking

**Q2:** When was the Latin version of the word Norman first recorded?

**A2:** 9th century

**Q3:** When was the French version of the word Norman first recorded?

**A3:** No Answers.

**Figure 1.** Question–answer pairs of SQuAD2.0. In this dataset, except the answer is No answer, other answers can be found in the passage.

The prevalent methods based on traditional deep learning, such as BiDAF [4], AoA-attention [5], R-Net [6], and QANet [7], use the static word vectors GloVe [8] or word2vec [9] as word representations and multilayer neural networks to enhance the model's understanding of the text and questions through continuous iteration and combine various attention mechanisms to enhance question-text relevance. However, the traditional word vector cannot address the problem of multiple meanings of a word. In addition, the attention module has a complex network structure, which makes it challenging for it to handle longer texts and causes several other problems, resulting in unsatisfactory MRC results.

Owing to the development of language models, the significant advances achieved in common language models can be used for various tasks [10–13], and surprising results have been achieved in MRC tasks. Yu et al. [14] used convolution kernels of different sizes to convolute and pool the encoding of BERT, and used global information and local information to fuse to improve the accuracy of reading comprehension of Chinese datasets. Pre-training language models (PrLMs) use the concept of transfer learning to train models effectively on a large corpus of relevant tasks. In addition, the parameters are fine-tuned in terms of a specific task to further optimize the models. However, based on the structural limitation of a model, the text needs to be truncated based on the maximum text length acceptable by it, leading to the loss of semantics. Further, such models are also unable to learn long-distance semantic relationships between sentences, which leads to their inability to understand text and related questions accurately.

To accurately represent the text, several researchers have focused on semantics and syntax [15–18]. SG-Net [19] explicitly considers syntactically significant words in each input sentence and selectively picks out such words to reduce the impact of noise caused by lengthy sentences. Parsing-All [16] benefits from each other with syntax and semantics as joint optimization goals. Zhang et al. [20] propose an approach based on semantic parsing to answer simple and complex questions and resolves ambiguity in natural language problems. Thus, a significant improvement has been achieved in syntactic parsing. Meanwhile, graph neural networks are beginning to be used for natural language task processing owing to their ability to model non-Euclidean spatial data. Fan et al. [21] used a combination of dependent syntax and graph convolution neural networks to perform a sentiment analysis based on comments from Internet users. Yin et al. [22] constructed parallel GCNNs and fused them with LSTM to extract graph domain features from feature cubes.

Zheng et al. [23] modeled text hierarchically at multiple-granularity levels and used graph attention networks to obtain various granularity representations to model the dependencies between the different granularities. Wu et al. [24] propose a novel Hierarchical-Cumulative Graph Convolutional Network (HC-GCN) to generating Social Relationship Graphs for Multiple Characters in Videos

To make up for the inability of the MRC model to deal with the two shortcomings of long texts and intersentence semantics, such that the model can understand long texts and accurately discriminate the semantics just like humans, we propose a graph neural network to feature textual sentences and in-sentence entities. Based on the feature that the different levels of granularity contain different levels of semantic information, a vast variety of semantic information is integrated to make the model's understanding of the various semantics accurate. Thus, whether a question is answerable or not can be effectively distinguished based on the granularity of the sentence. The start and end positions, which are immensely difficult to identify, can be obtained following multigranularity fusion. Experiments conducted on the SQuAD indicate that multigranularity syntax guidance (MgSG) outperforms traditional models in terms of both exact match (EM) and F score(F1).

The main contributions of this study are as follows:

- A new network structure, MgSG, is proposed. Based on the use of PrLMs to represent the text, combined with the graph structure, word and sentence granularities are used to obtain a text representation with richer semantics.
- Two graph structure construction methods are designed using dependencies and named entities, and a filtering mechanism is proposed to integrate them to improve the accuracy of the overall text representation.
- The role of the dependencies and named entities in reading comprehension tasks is analyzed, and it is demonstrated through experiments that both word and sentence granularities affect model performance. In addition, the two granularity representations are modularized to make them compatible with more models.

The proposed method is used for evaluation on the SQuAD and superior results are achieved in terms of both EM and F1 values when compared with the BERT-based reading comprehension model, demonstrating the effectiveness of the method.

The rest of the paper is organized as follows. Related work is summarized in Section 2. The various components of the model proposed in the text are described in detail in Section 3. The experimental results and analyses are presented in Section 4. The analysis and discussion of the effectiveness of the proposed method based on experiments are presented in Section 5. The conclusions are presented in Section 6.

## 2. Related Work

### 2.1. Machine Reading Comprehension

In recent years, span-extraction MRC has gained significant momentum [25–27]. It is a common practice to combine the two separate sequences of text and questions into one sequence in a particular way, and the attention mechanism plays a significant role in this. Zhu et al. [28] proposed the DUMA attention mechanism to directly model the MRC relationship as an attention mechanism that can effectively capture the relationship among the text paragraph, question, and answer triads. Zhuang et al. [29] designed DynSAN to enhance the model's ability to extract local semantics using a gated token selection mechanism to dynamically extract significant tokens from the sequence.

Graph neural networks have demonstrated unique advantages in NLP tasks [30–32]. The relational graph attention network (R-GAT) [33] proposed by Wang et al. addresses the problem of confusing connections when the model connects aspects with opinion words by encoding grammatical information. Ding et al. [34] proposed a CogQA framework based on multi-hop questions and answers in web-scale documents. Nicola et al. [35] focused their answer-seeking concerns on the integration and reasoning of information propagated within and between documents. The training and inference were performed using GCN. Yu et al. [36] analyzed the relationships among multiple documents and

queries, which was used to propose the bidirectional attention entity graph convolution network (BAG). The BAG uses the relationships between nodes in the entity graph and the attention information between the query and the entity graph to solve multi-hop reasoning QA task. Bhargav et al. [37] designed a deep neural architecture (TAP) for identifying answers and evidence in RCQA tasks requiring multi-hop inference. TAP consists of two loosely coupled networks: a local and global interaction extractor (LoGIX) and an answer predictor (AP). The loose coupling between the LoGIX and AP reveals the set of sentences used by the AP in predicting answers. Thus, the answer prediction of the TAP can be interpreted in a semi-transparent manner.

*2.2. Syntactic Representation*

Dependent syntax refers to the syntactic collocation between words. Recently, dependency syntactic parsing has been further developed using neural networks, and new state-of-the-art results have been obtained [38,39]. Benefiting from highly accurate parsers, neural network models can achieve higher accuracy by exploiting syntactic information rather than by ignoring it [15,40,41]. Kasai et al. [42] incorporated parse tree information by converting dependency labels into vectors and simply linking label embedding to word representations. Strubell et al. [43] proposed a neural network (LISA) that combines multiheaded self-attentiveness with multitask learning across dependency parsing, lexical tagging, predicate detection, and SRL. LISA encodes the sequences once for multiple lexical annotations and merges the syntax by training an attention head to attend to the grammatical parent of each token. Jawahar et al. [44] explored the layers of the BERT model and found that the lower, middle, and upper layers capture the surface, syntactic, and semantic features, respectively. The upper layer was found to model long-distance dependencies, making it critical to the performance of downstream tasks. However, they also found that syntactic information was diluted in the upper layers. Kuncoro et al. [45] extended the BERT model to consider syntactic information by modifying their pre-training target. They used an alternative syntactic language model as a learning signal, adding what they called "syntactic bias" to the BERT model. Using dependency trees and graph convolution networks to learn grammar-based embedding, Vashishth et al. [46] proposed SynGCN to overcome the problem of vocabulary explosion that arises when using the sequential context of words to learn word-embedding representations.

This study demonstrates the effectiveness of grammar and entity and graph neural networks in MRC when they accurately aid traditional BERT models in terms of embedding representations.

**3. Methodology**

In this section, we will focus on MgSG. Figure 2 depicts the overall architecture of MgSG, which consists of four parts: the text input, feature encoding, feature interaction, and answer prediction phases. We focus on and improve the performances of the feature encoding and feature interaction phases. In the feature encoding phase, text C and question Q are passed to the BERT model as the overall information <C, Q>. The syntactic parser is also used to syntactically parse <P, Q> and combine the sentence dependencies as features with the graph attention network. The resulting node-embedding representations are fused with the feature vectors generated by the BERT model and passed to the feature fusion module. In the fusion phase, the named entities in <P, Q> are passed to the graph neural network to reinforce their critical role in the text and answers, node-embedding representations are generated and passed to the attention network to update the embedding representations of the fused named entities, and finally the multiple-feature information is fused and passed to the model prediction phase.

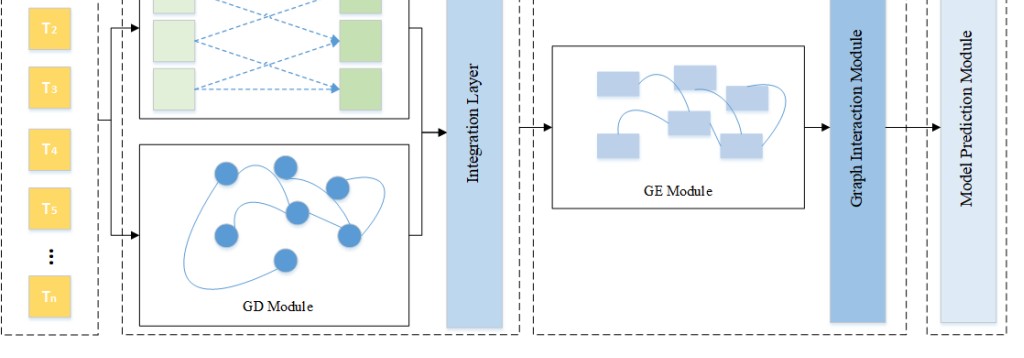

**Figure 2.** This is the overall structure of the model framework. The text is fed into the pretrained language model and our GD module (lower middle), respectively. In the GD module, a graph network is constructed according to the dependencies of the input text and fused with the output of pretrained language model. The GE module (middle right) extracts named entities in the text and builds a graph network and fuses it with BiDAF in the next layer. The model prediction module makes span predictions.

### 3.1. Feature Encoder Part

In this study, we input the text and questions as comprehensive information to the BERT model and graph with dependence (GD) module. The long-range semantic associations established by the graph structure are used to capture the semantic information of the text better, yielding a text-embedding representation that contains richer semantic information.

BERT Encoder

As presented on the left side of Figure 3, we concatenate the question and passage texts to form a single input sequence. In particular, we tokenize the input sequence to form word pieces (subword tokens) at first. Let C = $(c_1, c_2, \ldots, c_m)$ and Q = $(q_1, q_2, \ldots, q_n)$ denote the passage and question sequences of subword tokens of lengths $m$ and $n$, respectively. Let T = $([CLS], c_1, c_2, \ldots, c_m, [SEP], q_1, q_2, \ldots, q_n, [SEP])$ denote the total input sequence for the PrLM of subword tokens of length $L$, where $L = m + n + 3$. For each token, the input embedding is the sum of its token, position, and segment embeddings. Let X = $(x_1, x_2, \ldots, x_l)$ be the output from the embedding layer, which denote the embedding features of the input sequence tokens of length $l$. X is then fed to the BERT encoder to obtain a contextual representation. Let L = $(l_1, l_2, \ldots, l_l)$ be the sequence of outputs from the BERT model, which denote the embedding features of the sentence tokens of encoding length $l$. The outputs of BERT are particularly implemented as expressed in Equations (1) and (2):

$$X = Embedding(T) \tag{1}$$

$$L = Bert(X) \tag{2}$$

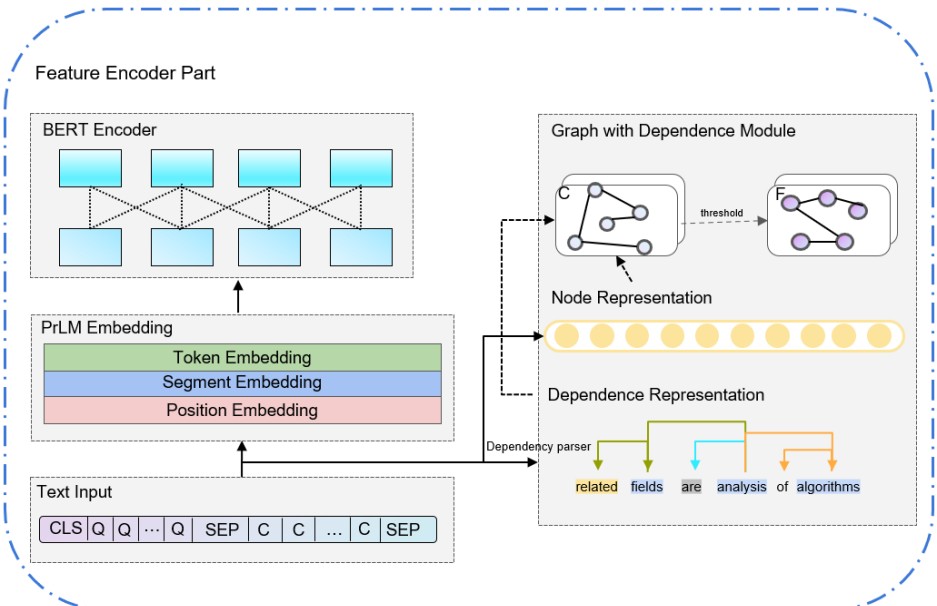

**Figure 3.** This is the overall structure of the feature encoder part. On the left side of the figure, PrLM applies special delimiters ((CLS) and (SEP)) to text sequences. On the right side of the figure, the GD module extracts the dependencies and builds the graph network.

### 3.2. Graph with Dependence Module

As presented on the right side of Figure 3, in the feature encoder phase, the long-range semantic associations established by the graph structure are used to better access the semantic information of the text. The syntactic parser (described in detail in Section 4) is also used to disassemble and analyze the text to generate a dependency graph and pass the dependency information to the graph attention network as features [47].

### 3.2.1. Dependence Relation

As can be seen from the bottom right of Figure 3, in this study, we use a pre-trained syntactic parser for the text with tokens as nodes to create dependency relations between each pair of nodes, that is, a syntactic dependency of interest (SDOI), by regarding each word as a child node; the SDOI consists of the child node and all its ancestor nodes in the dependency parsing tree [19]. That is, to exploit the relationship between the root word and the dependency of the dependency graph of the sentence, we need only focus on words that have a significant impact on the syntax. For example, in Figure 3, "related" does not have a impact on all the words but only on its ancestors "fields" and "analysis" and itself.

### 3.2.2. Graph Attention Network

We apply graph attention networks to model the information flow between nodes, which can further improve the representations of nodes through the attention mechanism over features from its neighbors. It can be seen from Figure 4 that the traditional one-dimensional convolution usually processes a few words near the central word, whereas the graph convolution considers these dependencies as edges. The central word and all other nodes connected by the edges are processed during the convolution. For example, the size of the convolution kernel depicted in Figure 4a is 3, and the process of convolution involves *related*, *fields*, and *are*. However, the graph convolution depicted in Figure 4b involves only *related* and *fields*.

In particular, the graph of GD module can be represented as $G(N, S)$, where $N$ is the set of nodes of the graph, $k$ is the number of nodes, $S$ is the set of edges of the graph, and an edge exists between two nodes if they are related. Figure 4 indicates that an undirected

graph can be constructed for all the tokens using the tokens as the graph's nodes and the edges of the graph as the association relations between the tokens.

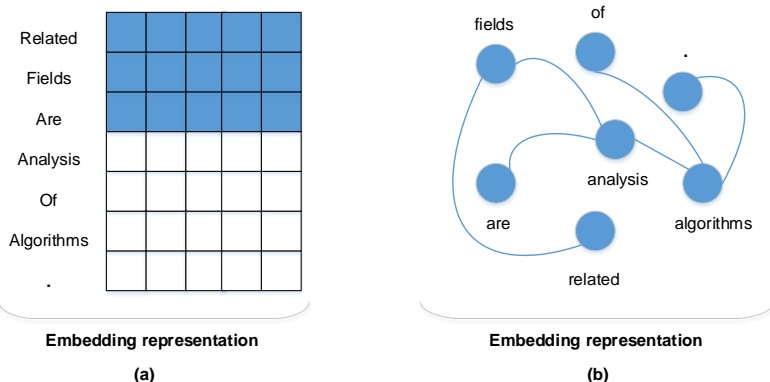

**Figure 4.** Difference between traditional convolution and graph convolution. (**a**) is the traditional convolution. (**b**) is the graph convolution.

In this study, We know that the graph node needs to use the word vector as a substitute for the word. GloVe [8] is used as the embedding representation for each token as a feature of the nodes; sentence-based dependencies are used as association relations. If the current token is not in GloVe's vocabulary, we will skip the current token and not act as a node. First, in the graph with dependency module, our input will generate multiple sets of dependencies after dependency analysis, and each set has multiple words with hierarchical relationships. Furthermore, the input is passed into the dependency parser, and if there are dependencies among the tokens, an edge weight initialized to 1 is added to both the nodes and 1 is added to the edge weight if the association is repeated subsequently. We also set a threshold to limit our filter on the level of dependency syntactic relations. If the number of dependencies set does not exceed the threshold, the information will not be incorporated into the graph structure. The edges of the graph are created as follows:

$$B_{dp} = \sum_{i=0}^{z} B_{dp}^{(i)} \tag{3}$$

where $B_{dp}$ represent the edges of the dependence relation, $z$ is the number of layers of dependencies. For example, extending the child node only depends on the parent node to the child node also depends on the parent node of the parent node, that is, two layers of dependencies.

Instead of other graph-based models, graph attention networks are used to maintain consistency with the multihead attention module of the BERT model. In the following, we will describe a single layer of the proposed graph attention network. For consistency with the multiheaded attention module in the BERT model, the graph attention network used in the text uses a multiheaded attention mechanism. We describe one of the $m$ attention heads here and $h_i$ is the representation of each node. All the parameters for each attention head and layer are unique. If an edge exists from node $j$ to node $i$, the attention factor $e_{ij}$ is calculated as follows:

$$e_{ij} = \frac{\left(h_i \mathbf{W}^{\mathrm{Q}}\right) \left(h_j \mathbf{W}^{\mathrm{K}}\right)^{\mathrm{T}}}{\sqrt{d_z}} \tag{4}$$

In the preceding equations, $\mathbf{W}^{\mathrm{Q}}$, $\mathbf{W}^{\mathrm{K}}$, and $\mathbf{W}^{\mathrm{V}}$ are parameter matrices.

We use the softmax function across all the neighbor nodes $j \in \mathcal{N}_i$ to normalize the

attention coefficients of node $i$. In particular, there is a self-loop for each node (that is, $i \in \mathcal{N}_i$) to allow it to update the feature. This process can be expressed as follows:

$$\alpha_{ij} = \text{softmax}_j\left(e_{ij}\right) = \frac{\exp\left(e_{ij}\right)}{\sum_{k \in \mathcal{N}_i} \exp(e_{ik})} \tag{5}$$

Further, the output of this attention head $z_i$ is computed as a weighted sum of the linearly transformed input elements:

$$z_i = \sum_{j \in \mathcal{N}_i} \alpha_{ij} h_j \mathbf{W}^V \tag{6}$$

where $d_Z$ is the output size of one attention head; we use $d_z \times m = d_h$. Finally, we concatenate the outputs of $m$ individual attention heads to obtain the multihead attention result $U \in \mathbb{R}^{d_h}$:

$$U = \|_{k=1}^{m} z_i^k \tag{7}$$

### 3.3. Integration Layer

The integration layer fuses the node- and text-embedding representations of the updated graph; this process is implemented using a multiheaded attention mechanism in a manner consistent with that explained in Section 3.2. Let M $= [m_1, m_2, \ldots, m_l]$ denote the output sequence for the integration layer of subword tokens of length $l$.

Referring to Zhang et al. [32], we use cross-attention as an additional layer to extract the answers to the questions to improve the accuracy of the prediction layer. We split the obtained sequences L and U into $L_Q$, $L_P$, $U_Q$, and $U_P$, where $L_Q$ and $L_P$ denote the representation parts of the question and text in L, respectively. $U_Q$ and $U_P$ have the same meaning as L. For dimensional unity, we expanded the split parts up to the maximum dimension.

$$\begin{bmatrix} q \\ k \\ v \end{bmatrix} = \begin{bmatrix} w_q \\ w_k \\ w_v \end{bmatrix} \cdot [L, U] \tag{8}$$

$$M = \text{multihead cross attention}\,(q, k, v) \tag{9}$$

### 3.4. Feature Interaction Part

The feature interaction module fuses the node and text vectors of the updated graph and is implemented using a multiheaded attention mechanism in a manner consistent with that explained in Section 3.3. The specific details are presented in Figure 5.

#### 3.4.1. Graph with Entity Module

The first step in initializing the graph structure is referred to as entity extraction. Named entities are vital for MRC. Considering the SQuAD as an example, we found that at least 95% of the questions or answers contained at least one entity through statistics on the dataset. Most of the remaining parts are time, number type entities, that is why we expand the scope of the entity. According to the characteristics of the dataset, we have extended the scope of named entities to include numeric entities and temporal entities. The range of named entities was extended to ensure the accuracy of the reading comprehension answers. Further, the keywords of a text are generally referred to as entities. As presented in Figure 6, the words typed in red-colored font are answer-related entities. It is clear to see that not only is the central idea of a passage usually the one that contains named entities but also these entities play a significant role in the answers. The fact that these entities appear several times at different locations, some of them at a distance from each other, provides us with ideas for using the graph structure to obtain semantics.

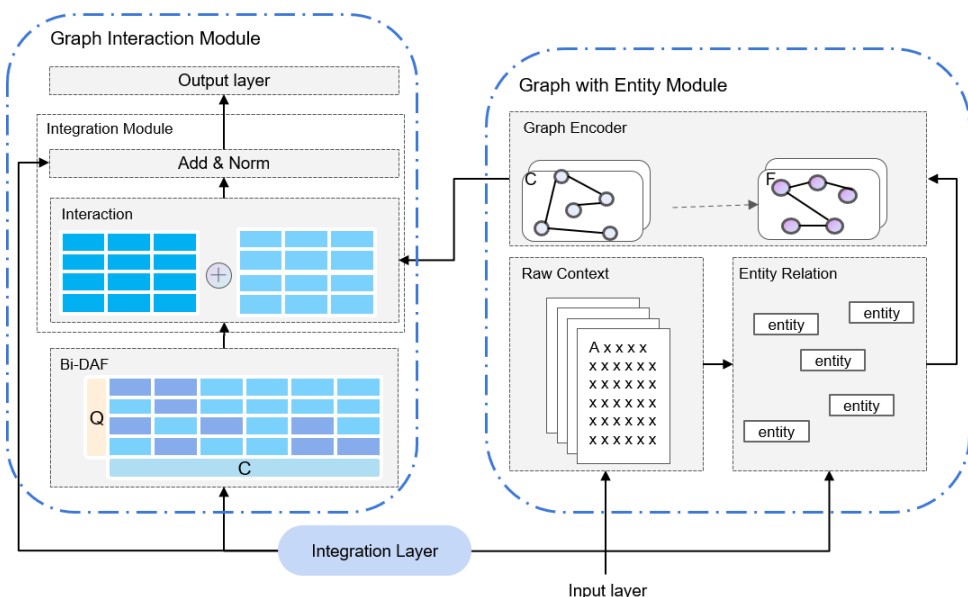

**Figure 5.** An overall architecture of the feature interaction module. On the right side of the figure, the GE module extracts named entities and builds a graph network based on specific relationships. Combining Bi-DAF (left side of the figure) with the output of GE for the segment prediction task.

Closely related fields in theoretical computer science are analysis of algorithms and computability theory. A key distinction between analysis of algorithms and computational complexity theory is that the former is devoted to analyzing the amount of resources needed by a particular algorithm to solve a problem, whereas the latter asks a more general question about all possible algorithms that could be used to solve the same problem. In turn, imposing restrictions on the available resources is what distinguishes computational complexity from computability theory: the latter theory asks what kind of problems can, in principle, be solved algorithmically.

**Figure 6.** Text content display.

In particular, we use the named entities obtained by extraction as nodes of the undirected graph, the embedding representation M output by the fusion layer mentioned in Section 3.3 as the feature vector of the named entities, and the relationship of association between the named entities as the edge weights. Here, we use the sliding window approach to obtain the edge weights: first, we define a sliding window $w$ and sliding step $s$, where $s < w$. After each text segment is counted, the sliding window will move backwards. If two or more named entities appear in a window $w$, an edge is attached to the nodes and initialized with weight 1. In the process of sliding the window, the content of the text may appear repeatedly, and we still use the same approach to calculate the weights. According to the characteristics of the dataset, we have extended the scope of named entities to include numeric entities and temporal entities. The range of named entities was extended to ensure the generality of the reading comprehension answers. Finally, we obtain an updated graph node representation G = ($g_1$, $g_2$, . . . , $g_n$).

3.4.2. Graph Interaction Module

The input of the fusion layer results, which is fed into the graph interaction module, is also fed into the Bi-DAF to obtain a contextual representation related to the query through

bidirectional attention and attention flow. Let $B = (b_1, b_2, \ldots, b_n)$ denote the outputs of Bi-DAF.

Up to this point, for each character in the text, an embedding representation $B$ based on the structure of the sequence can be obtained, where part of the named entities can also be represented by an embedding based on the graph structure $E$ The embedding representations obtained from the two structures are fused, resulting in a unified embedding representation containing the structures of both the features. The particular embedding representation is expressed as follows:

$$E(a_i) = \begin{cases} B(a_i), & a_i \notin e \\ B(a_i) + G(e_j), & a_i \in e_j \end{cases} \tag{10}$$

where $a_i$ is the token for which an embedding representation is required and $e$ is the set of named entity containing the token. At this point, the embedded representation information of the character is obtained from the sum of the embedded representations based on the sequence and graph structures. However, due to the BERT tokenize will cause the token to not correspond to the word, if the token $a_i$ does not belong to any of the named entities, then its embedding representation is obtained from the sequence-based textual embedding representation.

When the fused embedding representation is obtained, the residuals are concatenated with the output M of the initial fusion layer and regularized before being output to the model prediction module.

### 3.5. Prediction Module

The prediction module uses the text representations M and E based on the encoding and interaction modules to predict the answers to the questions. First, a residual connection is established between the two representations to avoid the gradient disappearing after multiple iterations, followed by layer normalization. We also utilize an additional fully connected feed-forward network. Finally, the Softmax function is used to predict the probability that a particular text is the answer. The specific calculations are as follows:

$$H = \text{LayerNorm}\,(\boldsymbol{M} + \boldsymbol{E}) \tag{11}$$

$$P_a = \text{Softmax}(\boldsymbol{W}_a\boldsymbol{H} + \boldsymbol{b}_a) \tag{12}$$

## 4. Experiment

In this section, we introduce our experiments in detail, including the experimental settings, datasets, evaluation metrics.

### 4.1. Setup

For the syntactic parser, we adopt the dependency parser from dependency parser (HPSG) [48] through joint learning of the constituent parsing using the BERT model as the sole input, which achieves very high accuracy: 97.00% UAS and 95.43% LAS on the English dataset Penn Treebank (PTB) test set.

We use the available PrLMs as encoders to build the baseline MRC models: BERT and ALBERT. Our implementations of the BERT and ALBERT models are based on the public Pytorch implementation from Transformers. We use the PrLM weights in the encoder module and all the official hyperparameters. For the fine-tuning, we set the initial learning rate within $\{2 \times 10^{-5}, 3 \times 10^{-5}\}$ with a warm-up rate of 0.1 and L2 weight decay of 0.01. The batch size is selected from within $\{8, 16, 32\}$. The maximum number of epochs is set to 4 for all the experiments. The hyperparameters are selected using the dev set.

### 4.2. Datasets

Our proposed model is evaluated on two MRC datasets: SQuAD2.0 and CMRC 2018 [49].

SQuAD2.0 is an MRC dataset on Wikipedia articles with more than 150K questions. As a widely used MRC benchmark dataset, SQuAD2.0 is an upgrade of SQuAD1.1 with over 50,000 new and unanswerable questions that are written adversarially by crowdworkers to look similar to answerable ones. The training dataset contains 87 K answerable and 43 K unanswerable questions.

CMRC is a span-extraction dataset for Chinese MRC. This dataset consists of nearly 20,000 real questions annotated in Wikipedia passages by human experts. Before annotation, a document is divided into several articles, and each article is no more than 500 words in length.

### 4.3. Evaluation

The EM and F1 are used to evaluate the performance of the model at the token level. EM denotes the percentage of answers predicted by the model in the dataset that are the same as the ground truth. F1 denotes the average word coverage between the model-predicted answers and ground truths in the dataset.

## 5. Results and Discussion

In this section, we demonstrate the experiment results and analyze the reasons for the results.

### 5.1. Experiment Results

We present a comparison between the proposed model and the baseline models on the SQuAD 2.0 and CMRC 2018 dev sets, including three traditional methods and several fine-tuned methods based on PrLMs. The main results are presented in Tables 1 and 2.

**Table 1.** Results for SQuAD 2.0 dev dataset. The results of the experiments are obtained in our local environment and the corresponding literatures.

| Model | EM(%) | F1(%) | EM(%) for MgSG | F1(%) for MgSG |
|---|---|---|---|---|
| Human performance | 86.8 | 89.5 | | |
| QANet | 62.6 | 66.7 | | |
| SAN | 68.2 | 70.9 | | |
| Match-LSTM | 60.3 | 63.5 | | |
| BERT-base | 75.8 | 79.2 | 77.2 | 81.1 |
| BERT-large | 80.4 | 83.3 | 80.9 | 84.0 |
| ALBERT-base | 77.1 | 80.1 | 81.5 | 84.8 |
| ALBERT-large | 79.4 | 82.3 | 81.8 | 85.6 |
| ALBERT-xxlarge | 85.6 | 88.1 | 85.9 | 88.2 |
| XLNET-base | 77.6 | 80.3 | 79.5 | 82.1 |
| PERT-base | 76.3 | 80.1 | 76.9 | 80.5 |
| PERT-large | 82.8 | 86.1 | 82.5 | 86.0 |
| ELECTRA-base | 79.5 | 82.5 | 80.2 | 83.6 |

The results indicate that the proposed model combined with the various pre-trained models mentioned indicate varying degrees of improvement, some of which are significant. For the SQuAD dataset, comparing the same pre-trained models but with different sizes. Although the improvement of the large model is not as large as that of the small model, there is still a relatively obvious improvement. For example, the EM of BERT-large has increased by 0.5%. Compared with the large improvement of the small model, because the learning ability of the large model is strong enough, some grammatical relations have been learned, which leads to the fact that the additional feature information we have may have been learned, so compared with the small model, the improvement is not very obvious. Then, the improvement for albert-base+MgSG is better than that of BERT-large,

and the amount of parameters is much smaller than that of BERT-large. This proves that the improvement of our module is more obvious for small models, which provides a new idea for model Lightweighting.

**Table 2.** Results for CMRC-2018 dev dataset. The results of the experiments are obtained in our local environment and the corresponding literatures.

| Model | EM(%) | F1(%) | EM(%) for MgSG | F1(%) for MgSG |
|---|---|---|---|---|
| Human performance | 75.8 | 79.2 | | |
| P-Reader (single model) | 76.7 | 80.6 | | |
| Z-Reader (single model) | 76.3 | 80.5 | | |
| BERT-base | 63.6 | 83.9 | 64.3 | 84.4 |
| BERT-wwm-ext | 64.6 | 84.8 | 64.8 | 84.8 |
| RoBERTa-wwm-ext | 65.5 | 85.5 | 65.1 | 85.6 |
| ELECTRA-base | 66.9 | 83.5 | 67.5 | 84.7 |
| ELECTRA-large | 67.6 | 83.8 | 67.9 | 85.4 |
| PERT-base | 64.1 | 84.5 | 64.8 | 85.1 |
| MacBERT-base | 66.3 | 85.4 | 66.1 | 85.4 |
| ERNIE 2.0-base | 67.8 | 87.5 | 67.7 | 87.3 |

For the CMRC dataset, it can be seen from Table 2 that the fusion of multigranularity features with the baseline model also effects a performance improvement of approximately 1% over the baseline model. However, CMRC is a Chinese dataset. Different languages bring different grammatical rules and semantics. Compared with Chinese grammar rules, English grammar rules are obviously clearer, and the semantic information contained in the dependencies of words is also clearer, which can greatly help the understanding of sentences. Therefore, the improvement of the model for English data is greater. Due to the grammatical structure of Chinese is more random and irregular, which makes it difficult for us to use the information contained in Chinese grammar accurately and correctly. On the other hand, it proves that our model can make good use of the hidden information contained in explicit grammatical structures and the use of weak grammatical structures is less stable.

*5.2. Ablation Study*

In this section, we describe certain ablation experiments that were conducted to demonstrate the validity of the proposed model. All the experimental baselines refer to the BERT base model except for the comparison of changes in variable.

To verify the effectiveness of our module, we separately added two modules to the baseline model. From the experimental results presented in Table 3, we can see that when we add the GD module and the graph with entity (GE) module to the baseline, the performance of the model is significantly improved. Further, combined with the results presented in Table 1, we understand that these two modules can improve performance when used alone, and they are compatible and improve each other's performance when used together.

**Table 3.** Impact of various modules on the performance of the model.

| Model | EM (%) | F1 (%) |
|---|---|---|
| BERT | 75.8 (±0.2) | 79.2 (±0.2) |
| BERT + GD | 76.7 (±0.1) | 80.6 (±0.2) |
| BERT + GE | 76.5 (±0.1) | 80.5 (±0.2) |
| BERT + GD + GE | 77.2 (±0.1) | 81.1 (±0.2) |

We also demonstrate the effectiveness of our proposed method with an example. As shown in Figure 7, in the first example, the question is *What chemical did Priestley use in his experiments on oxygen?* and the answer is obviously in the vicinity of the person named entity in the question in the paragraph, and then the BERT model. The answer was not predicted, and the answer suggested by our model was mercuric oxide. This shows that by constructing part-of-speech edges, the model can have a more specific goal in finding answers. The problem of the second example in Figure 7 is *Which is one of the park features located in North Fresno?*, the difficulty of this question is that it does not provide a very direct keyword, *North Fresno, feature* can only provide the approximate location of the answer in the text and there are multiple misleading answers around. Our method can give the correct answer because it follows the grammatical structure and reinforces the hidden information contained in *Woodward Park* as the subject of *which features the Shinzen Japanese Gardens*, while BERT cannot.

We also verify the effectiveness of the dependency syntax through experiments. We will first average the hidden state representations of the last layer of BERT and generate the corresponding heatmap (Figure 8). We find that BERT is significantly more interested in regular subject position, but this is not where the answer comes from. We then summed and averaged the output of our module with the representation of the hidden state. The darker the color, the higher the impact. Due to the addition of dependency syntax, the attention of the model significantly weakens the position of the beginning, and is biased towards the parenthesis in the sentence, which is the source of the correct answer.

In the meantime, on August 1, 1774, an experiment conducted by the British clergyman Joseph Priestley focused sunlight on mercuric oxide (HgO) inside a glass tube, which liberated a gas he named "dephlogisticated air". He noted that candles burned brighter in the gas and that a mouse was more active and lived longer while breathing it. After breathing the gas himself, he wrote: "The feeling of it to my lungs was not sensibly different from that of common air, but I fancied that my breast felt peculiarly light and easy for some time afterwards
What chemical did Priestley use in his experiments on oxygen?
BERT answer：None          BERT+MgSG answer：mercuric oxide ( √ )

Fresno has three large public parks, two in the city limits and one in county land to the southwest. Woodward Park, which features the Shinzen Japanese Gardens, numerous picnic areas and several miles of trails, is in North Fresno and is adjacent to the San Joaquin River Parkway. Roeding Park, near Downtown Fresno, is home to the Fresno Chaffee Zoo, and Rotary Storyland and Playland. Kearney Park is the largest of the Fresno region's park system and is home to historic Kearney Mansion and plays host to the annual Civil War Revisited, the largest reenactment of the Civil War in the west coast of the U.S.
Which is one of the park features located in North Fresno?
BERT answer：Woodward Park          BERT+MgSG answer：Shinzen Japanese Gardens( √ )

**Figure 7.** The red font represents the correct answer, blue font represents the wrong answer, green fonts represent keywords or information in the question.

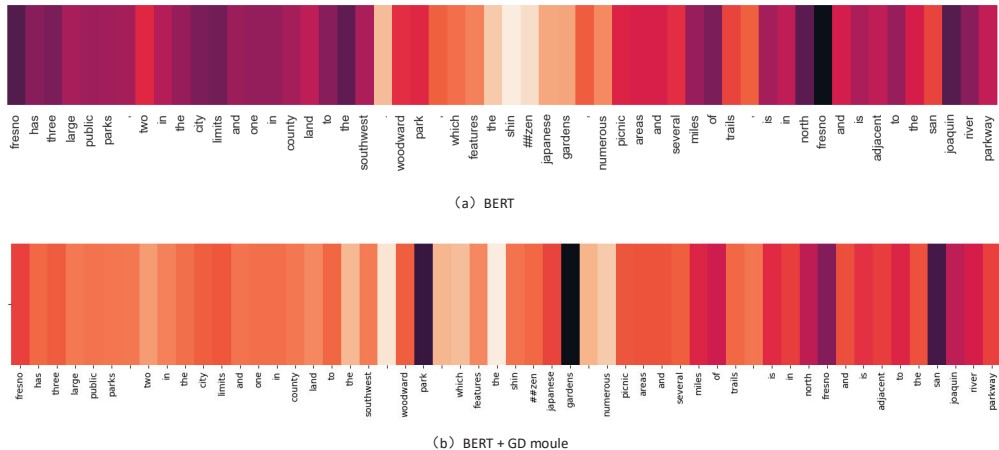

**Figure 8.** Effectiveness analysis. The darker the color of the token, the greater the influence of the token.

Further, as illustrated in Figures 9 and 10, we present two different examples using the SQuAD dataset to demonstrate the effectiveness of the proposed method. For the example presented in Figure 9, the answer is a name. In addition, not too many other named entities are present in this sentence. Therefore, the named entity granularity effectively reinforces the role of the answer in this sentence. When only named entity granularity is added to the model, the probability of predicting an answer improves by almost 2%. However, when the only dependency granularity is added, model is 1.6% less likely to predict an answer. The reason for such results is that the syntactic complexity of the sentence is too high because it contains multiple clauses, causing the model to focus too much on the relationship between the sentences and not pay attention to the answer. When we add both granularities to the model, it still has a positive effect on the prediction of the answer. This is consistent with the data presented in Table 1.

**Passage:** The Duchy of Normandy, which they formed by treaty with the French crown, was a great fief of medieval France, and under Richard I of Normandy was forged into a cohesive and formidable principality in feudal tenure.

**Q:** Who ruled the duchy of Normandy?
**Ground-Truth:** Richard I
**Predication:** Richard I
**Probability:** 0.718

**+ Named Entity Granularity**
**Probability:** 0.734

**+ Dependency Granularity**
**Probability:** 0.702

**+ Named Entity Granularity and Dependency Granularity**
**Probability:** 0.725

**Figure 9.** Effectiveness analysis. Example of the answer is the possibility of named entities.

Figure 10 presents another form of the answer, which itself increases the difficulty of the model predicting the answer when the answer is a sentence. When there are a few named entities in the sentence, there is a high probability that they will occur in the answer. In this example, *solutions* and *resources* are reinforced. The dependency relationship plays a more significant role in this example, and this specific dependency relationship analysis is illustrated in Figure 7. The results obtained from these two examples demonstrate the effectiveness of the proposed method.

---

**Passage：** A problem is regarded as inherently difficult if its solution requires significant resources, whatever the algorithm used. The theory formalizes this intuition, by introducing mathematical models of computation to study these problems and quantifying the amount of resources needed to solve them, such as time and storage.

---

**Q：** What measure of a computational problem broadly defines the inherent difficulty of the solution?

**Ground-Truth：** if its solution requires significant resources

**Predication：** if its solution requires significant resources

**Probability：** 0.378

**+ Named Entity Granularity**

**Probability：** 0.394

**+ Dependency Granularity**

**Probability：** 0.409

**+ Named Entity Granularity and Dependency Granularity**

**Probability：** 0.415

**Figure 10.** Effectiveness analysis. Example of the answer is the possibility of long sentence.

*5.3. Discussion*

In this section, we present a deep analysis of the effect of extensions to named entities for the various parts of speech and multiple parameters with dependencies on the performance of the model.

First, we experimented with and analyzed the relationship between the number of layers and the performance of the graph neural network in which the dependencies were located. Further, to verify the effect of graph sparsity, we analyzed the values of the dependency threshold and the categories of the named entity. The experimental results are presented in Tables 4–6.

Table 4 indicates that as the number of dependent layers increased, the F1 value of the model steadily increased from layer 1 to 4. This result suggests that more interword relationships can be obtained by interfusing pairs of nodes at further layers, forming a better understanding of the sentences. However, as the number of layers increases to 5, the evaluation index decreases. Thus, the experimental results suggest that when the number of layers of dependencies increases to a certain number, it is counterproductive to learn interword relationships after the model has already learned them.

**Table 4.** Impact of the number of dependency layers on the performance of the model.

| Num of Layers | EM (%) | F1 (%) |
|:---:|:---:|:---:|
| 1 | 76.1 | 80.1 |
| 2 | 75.5 | 80.2 |
| 3 | 76.8 | 80.8 |
| 4 | 77.2 | 81.1 |
| 5 | 76.5 | 80.8 |
| 6 | 76.4 | 80.5 |

Further, as presented in Table 5, we analyze the effect of dependency thresholds on the performance of the model. We argue that when there are only a very few words in a sentence as a collection of subwords, the words in this collection should not represent the meaning of the sentence; a higher number of subwords are more representative of the meaning of the sentence. The experimental results also verify our conjecture that such a fusion will instead affect the performance of the model when the dependency threshold is

0, that is, when all the words have dependencies. When we increased the threshold to a certain value, we obtained the best balance.

**Table 5.** Impact of the threshold of GD on the performance of the model.

| Threshold | EM (%) | F1 (%) |
|---|---|---|
| 0 | 75.5 | 78.8 |
| 1 | 76.8 | 80.2 |
| 2 | 76.9 | 80.5 |
| 3 | 76.6 | 80.1 |
| 4 | 76.1 | 79.5 |

This experiment proves the correctness of our extended entity range. By adding different entity types, we found that although the expanded entity types do not account for a high proportion of the overall dataset, the numbers usually contain extremely high information in a piece of text, and even some answers are numbers. In addition, we also thought that some pronouns actually sometimes refer to some entities, and in some sentences, entities are omitted and pronouns are used. However, when we added the upper pronoun, the results were lower. The main reason is that the number of words that pronouns can refer to is too large, and the referential relationship is very difficult to handle. This leads to the reason for the performance degradation caused by adding pronouns.

**Table 6.** Impact of the POS on the performance of the model.

| Type | EM (%) | F1 (%) |
|---|---|---|
| Noun | 76.5 | 80.4 |
| Noun + Pron | 76.1 | 80.2 |
| Noun + Time | 76.6 | 80.5 |
| Noun + Num | 76.7 | 80.5 |
| Noun + Num + Time | 76.7 | 80.7 |

## 6. Conclusions

This paper proposes a MRC model that combines graph neural networks with multiple-granularity semantic fusion. The model takes advantage of the graph network structure that can link long-distance nodes using sentence dependencies and entities as two important features to help PrLM with a smaller number of parameters to obtain more accurate text and question representations by adding fewer parameters and computational cost. Using the BERT as the baseline model, MgSG achieves significantly better results than the baseline model on the SQuAD2.0, demonstrating that MgSG has a significant impact on MRC. At the same time, we also found that our method can better learn explicit grammatical structures or clear grammatical structures (such as English). This provides new ideas for future research on model lightweighting and analysis of syntactic semantics.

In the future, we will try to use other graph structures(i.e., graph-to-sequence models) to learn more features and delve into how to make the model better understand data with unclear grammatical structures, such as Chinese, to achieve a better understanding of the text.

**Author Contributions:** Conceptualization, Z.L. and G.L.; methodology, Z.L.; validation, Z.L., C.X., C.Z. and Y.Z.; formal analysis, Z.L.; investigation, Z.L.; resources, Z.L.; data curation, Z.L.; writing—original draft preparation, Z.L.; writing—review and editing, G.L., Z.L., C.X., C.Z. and Y.Z.; supervision, G.L., C.Z. and C.X.; project administration, G.L. and Z.L.; funding acquisition, G.L. and C.X. All authors have read and agreed to the published version of the manuscript.

**Funding:** This work was supported in part by the China Chongqing Science and Technology Commission under Grant cstc2020jscx-msxmX0086, cstc2019jscx-zdztzx0043, cstc2019jcyj-msxmX0442. China Chongqing Banan District Science and Technology Commission project under Grant 2020QC413, and

China Chongqing Municipal Education Commission under Grant KJQN202001137. Moreover, this work is Chongqing University of Technology Graduate Education Quality Development Action Plan Funding Results (Project number: gzlcx20223456).

**Institutional Review Board Statement:** Not applicable.

**Informed Consent Statement:** Not applicable.

**Data Availability Statement:** Data are contained within the article and anyone can be used.

**Conflicts of Interest:** The authors declare no conflict of interest.

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
