# Peer review of "Multigranularity Syntax Guidance with Graph Structure for Machine Reading Comprehension"

_applsci, doi:10.3390/app12199525_

Round 1
Reviewer 1 Report
The paper addresses the task of Machine reading comprehension by leveraging several Transformer-based, pretrained language models and graph neural networks. The authors used two datasets to evaluate the model: SQuAD 2.0 and CMRC-2018. Their model uses two modules, a feature encoder and feature interaction, the first one extracts not only the language model features but also through graph convolutions a representation of their syntactic dependencies. The second module combines the Bi-DAF model results with another graph convolutional network over the cooccurrences of the named entities in the question context.
Line 117 - "Nicola et al" is rather De Cao et al.
Line 185 - "As can be seen from the bottom right of Figure 3, in this study, we use a pre-trained syntactic parser" - unfortunately, this is not obvious in the figure
Line 185-188 - There might be useful to provide a link to the pretrained model
Chapter 3.2.2 - mentioning how Unknown tokens are treated in the context of GloVe Embedding would help the reproducibility of the paper.
Around Line 204 - What justifies the necessity to maintain the "consistency with the multiheaded attention module in the BERT model"? Is there any literature or self-conducted experiments that would prompt this necessity?
Line 217 - Please justify the assertion "Named entities are vital for MRC."
Chapter 3.4.1 - it is unclear how the NER is performed, what models were used and what types of entities are used for the Graph.
Line 312 - the performance of the model is significantly improved – please use a statistical method to test whether the differences are indeed significant. Nevertheless, for some models, there is a slight improvement and even a performance decrease. For CMRC-2018, half of the models under-performed the baselines
General Question about the Results - are the results computed as a mean over several runs (if this is the case, how many), or just one run? If the authors used the results from one run, how do they ensure that the result is consistent, and not the result of a "lucky random seed"
Around line 237 - "Up to this point, for each character in the text, an embedding representation B based on the structure of the sequence can be obtained" - As far as we understand, Bi-DAF contextualizes character and word embeddings and encodes the interaction among the context words conditioned by the query. Does the MgSG use only the character embeddings from Bi-DAF?
Regarding the model used for the two datasets, we wonder for CMRC-2018 if there is a specific reason for not using all of the baseline models in combination with MgSG as it was done for SQuAD (For instance Electra+MgSG is missing, etc). Additionally, if there is a specific reason for not testing ALBERT on CMRC-2018
Line 312 "The performance of the model is significantly improved.",
Table 3 - Please explain the decrease in performance in case of adding the GE module
Another major concern arises – the baseline with only BERT is too simple, SG-NET reported better performance 2+ years ago. Please refine your baseline and consider an in-depth discussion with better models
Tables 4 and 5 - The analysis of the impact of the hyperparameters stops after the first decrease in performance. To outline a downward trend, two data points would be much more convincing than one.
Table 6 – the differences seem marginal, please test whether they are statistically significant or indeed not. What about descriptive entities (i.e., adjectives and adverbs?)
The Conclusions section is too concise, a more in-depth presentation of the conclusions would be more advised. Also, these should address the lower performance on the CMRC-2018 dataset compared to SQuAD
Additionally, it is unclear how the paper addresses the issue mentioned on line 72 "overcoming the problem of understanding long texts"
Author Response
Thank you for the detailed review. We have carefully and thoroughly proofread the manuscript to correct all the grammar and typos. For some format, pictures, citations and other issues, we will mark the newly submitted documents, and will no longer make explanations in the area below to save your review time. We would like to take this opportunity to thank you for all your time involved and this great opportunity for us to improve the manuscript. We hope you will find this revised version satisfactory.
Reviewer#1, Concern # 1:mentioning how Unknown tokens are treated in the context of GloVe Embedding would help the reproducibility of the paper.
Author response: Thanks for your professional suggestion. We are very sorry that you cannot understand the usage of this part because we do not explain in detail. Our input data will be fed into BERT and graph neural network respectively (as shown in Figure 1). GloVe is used here as a word vector. When our selected word is used as a graph node, a digital matrix is required as the expression form of the word. Due to the advantage of GloVe being more global in training, we use it as a word vector. If the current token is not in the vocabulary of GloVe, we will skip the current token and not act as a node. We made changes around line 203 in the latest submitted manuscript.
“GloVe[ 8] is used as the embedding representation for each token as a feature of the nodes; sentence-based dependencies are used as association relations. If the current token is not in GloVe’s vocabulary, we will skip the current token and not act as a node.”
Figure 1
Reviewer#1, Concern # 2: What justifies the necessity to maintain the "consistency with the multiheaded attention module in the BERT model"? Is there any literature or self-conducted experiments that would prompt this necessity?
Author response: Thanks for your professional suggestion. We know that the multi-head attention mechanism divides the model into multiple heads to form multiple subspaces, allowing the model to focus on different aspects of information without increasing the computational complexity. At the same time, an experiment in the paper “What Does BERT Look At?” shows that the gap between heads decreases as the number of layers becomes larger. That is, the variance between heads decreases as the number of layers increases. The paper “Analyzing Multi-Head Self-Attention” shows that more heads are not always better. From this, our initial idea is to make the number of heads of the attention mechanism of different models consistent, so that the same text can be learned in the same number of subspaces, and the difference between the two models is as small as possible. Fortunately, most people do the same. We also verified our ideas through experiments. We found that when the number of GAT heads is similar to the number of BERT-base heads, the effect is better, but the fluctuation is small.
|
GAT_head_num |
EM(%) |
F1(%) |
|
8 |
77.1 |
81.1 |
|
12 |
77.2 |
81.1 |
|
16 |
77.0 |
81.0 |
Clark K, Khandelwal U, Levy O, et al. What does bert look at? an analysis of bert's attention[J]. arXiv preprint arXiv:1906.04341, 2019.
Voita E, Talbot D, Moiseev F, et al. Analyzing multi-head self-attention: Specialized heads do the heavy lifting, the rest can be pruned[J]. arXiv preprint arXiv:1905.09418, 2019.
Reviewer#1, Concern # 3:Please justify the assertion "Named entities are vital for MRC."
Author response: Thanks for your professional suggestion. First of all, as a kind of thinking instinct, in order to quickly understand the general idea of the article when we are reading comprehension, we usually choose to find some keywords first, and the parts of speech of keywords are generally nouns. Regular named entities are names of people, institutions, places, and all other entities identified by names. However, when we analyzed the dataset, we found that at least 95% of the questions or answers contained at least one entity through statistics on the SQuAD dataset. Most of the remaining parts are time, number type entities, that's why we expand the scope of the entity. There are also related additions around line 220 in the latest manuscript. At the same time, in about 225 lines of the article, we also use a small example to prove that the central idea of the article generally contains multiple entities or even one entity appears multiple times to highlight the main idea. It is clear that these entities appear multiple times in different locations, some of which are at a distance from each other, and this fact gives us the idea of using the graph structure to obtain semantics. At the same time, the data in Table 3 also proves that when we add entity relations, the model performance is improved.
Reviewer#1, Concern # 4:it is unclear how the NER is performed, what models were used and what types of entities are used for the Graph.
Author response: Thanks for your professional suggestion. We're sorry we didn't explain this clearly to you. As mentioned above, entities are very important for MRC, and there may be some gaps between entities, which makes us think that we can use the graph structure to solve this problem. First, we will extract the entities in the input information. After extracting named entities, construct the extracted named entities as an undirected graph. The vertices in this graph are the extracted named entities, and the edges in the graph are the associations between named entities. We use the co-occurrence relationship as the association between named entities: if two named entities appear at the same time in the same text segment, add an edge to the vertex corresponding to the two named entities. The selection of text fragments is explained around line 229 in the paper. For the specific entity type, we choose the noun entity, and at the same time, according to the characteristics of the machine reading comprehension task, we also add the number entity and the time entity into it. The specific experimental results have been shown in Table 3 in the manuscript.
Reviewer#1, Concern # 5:please use a statistical method to test whether the differences are indeed significant. Nevertheless, for some models, there is a slight improvement and even a performance decrease.
Author response: Thanks for your professional suggestion. As the most well-known public dataset, SQuAD is exciting enough for businesses and professionals all over the world to see even a 1% improvement without modifying pretrained language models. However, we choose to exploit the properties of graph structure, creatively use named entities and syntactic dependencies as features in the MRC task and achieve good improvements. It provides new features and new ideas for researchers to study this field in the future. In addition, as shown in the table below, the number of parameters added by the method proposed in this paper is negligible compared to the pre-trained language model with hundreds of millions of parameters, so the increase in inference time is almost nothing.
|
Model |
parameter |
inference time |
EM(%) |
F1(%) |
|
BERT-base |
110M |
164s |
75.8 |
79.2 |
|
BERT-base+MgSG |
113M |
182s |
77.2 |
81.8 |
|
BERT-large |
340M |
481s |
80.4 |
83.3 |
|
BERT-large+MgSG |
343M |
537s |
80.9 |
84.1 |
|
ALBERT-base |
11M |
176s |
77.1 |
80.1 |
|
ALBERT-base+MgSG |
14M |
193s |
81.5 |
84.8 |
Although the improvement of the large model is not as large as that of the small model, there is still a relatively obvious improvement. For example, the EM of BERT-large has increased by 0.5%. Compared with the large improvement of the small model, because the learning ability of the large model is strong enough and some grammatical relations have been learned, it proves that our additional dependencies may have been learned, so the improvement is not a problem compared to the small model. The improvement for ALBERT-base+MgSG is better than that of BERT -large, and the number of parameters is much smaller than that of BERT-large. This proves that our module improves significantly for small models.
We exploit word dependencies to facilitate semantic understanding. CMRC is a Chinese dataset. First, the research on pre-trained language models for Chinese is not as in-depth as the pre-trained language models for English, resulting in a large gap between the number and performance of Chinese models and English. Secondly, compared with the grammatical rules of Chinese, English is strict about the rules of grammar, the dependencies of words contain more semantic information that can facilitate the understanding of sentences. Therefore, the improvement of the model for the English data set is more obvious.
Finally, we attach a related paper of AAAI in 2020, the specific content is not introduced. The paper also uses the SQuAD dataset, and the improvement of SG-Net compared to the baseline is 1%, which is an exciting improvement. The improvement of our corresponding model is 1.4%, so the improvement in this paper is not insignificant. We made changes around line 304 in the latest submitted manuscript.
Zhang Z, Wu Y, Zhou J, et al. SG-Net: Syntax-guided machine reading comprehension[C]//Proceedings of the AAAI Conference on Artificial Intelligence. 2020, 34(05): 9636-9643.
Reviewer#1, Concern # 6:Regarding the model used for the two datasets, we wonder for CMRC-2018 if there is a specific reason for not using all of the baseline models in combination with MgSG as it was done for SQuAD (For instance Electra+MgSG is missing, etc). Additionally, if there is a specific reason for not testing ALBERT on CMRC-2018
Author response: Thanks for your professional suggestion. First of all, we apologize because the experimental data of ELECTRA-base+MgSG and ELECTRA-large+MgSG cannot be displayed in the table due to our mistakes. These two experimental data existed before the manuscript was submitted but were omitted due to personal negligence. We have added it in the latest manuscript. At the same time, because CMRC is a Chinese data set, and some pre-trained language models only support English input (such as ALBERT), there will be some differences between the models in the two data sets.
Reviewer#1, Concern # 7:Table 3 - Please explain the decrease in performance in case of adding the GE module.
Tables 4 and 5 - The analysis of the impact of the hyperparameters stops after the first decrease in performance. To outline a downward trend, two data points would be much more convincing than one.
Table 6 – the differences seem marginal, please test whether they are statistically significant or indeed not. What about descriptive entities (i.e., adjectives and adverbs?)
Author response: Thanks for your professional suggestion. We're sorry that we misunderstood you with our table. The +GE module in this table is not an addition based on the BERT+GD module. Instead, they are added separately to prove that both modules play a role in improving the performance of the model. So, the +GE module is not a performance degradation, it proves the correctness of our method anyway. Usually, named entities are names of people, institutions, places, and all other entities identified by names, and there are almost no adjectives and adverbs. The original purpose of this experiment was just to demonstrate that our extension entity was meaningful. We have also revised the representation of the table in the latest manuscript. The data in Table 4, Table 5 and Table 6 are also updated in the new version of the manuscript. Sorry again for your misunderstanding. We made changes around line 327 in the latest submitted manuscript.
Reviewer#1, Concern # 8:General Question about the Results - are the results computed as a mean over several runs (if this is the case, how many), or just one run? If the authors used the results from one run, how do they ensure that the result is consistent, and not the result of a "lucky random seed"
Author response: Thanks for your professional suggestion. The experimental data we provide are the average of two experimental data. We know that random numbers will affect the experiment during the running of the program, so we have limited seed for both numpy and torch.manual_seed. In order to prevent other random factors from affecting the effect, the same seed was selected for both experiments.
Reviewer#1, Concern # 9:the baseline with only BERT is too simple, SG-NET reported better performance 2+ years ago. Please refine your baseline and consider an in-depth discussion with better models
Author response: Thanks for your professional suggestion. Since other pre-trained language models are changed from BERT, and most of the papers use BERT as the Baseline, which is why we choose it as the Baseline. Please let me give an introduction to the differences between SG-Net and our proposed method. First of all, SG-Net also uses BERT as the baseline and adds grammatical structure information as features for improvement and experimentation. However, SG-Net is an innovation that combines grammatical structure information with an attention mechanism, which leads to this method requiring a large modification and re-pre-training of the original BERT model, which is costly in terms of time and computation. All are very expensive. While MgSG is similar to an external feature supplementation module, it can be used without extensive modification of BERT. Therefore, we believe that SG-Net is different from our method in the essential starting point, so it is not added to the table for data comparison. Secondly, the data on the paper of SG-Net shows that the EM value of SG-Net is 1% higher than the Baseline. However, the EM value of our BERT+MgSG is 1.4% higher than that of Baseline. The effect is better than SG-Net. Finally, the experimental results of the SG-Net article can also prove that the effect of our proposed method is not insignificant, but significant.
Reviewer#1, Concern # 10:Additionally, it is unclear how the paper addresses the issue mentioned on line 72 "overcoming the problem of understanding long texts"
Author response: Thanks for your professional suggestion. In theory, our MgSG module can accept text input of any length. Because this module only needs to extract the syntactic dependency structure and named entities in the input text as the nodes of the graph structure. At the same time, we will use the words in it as the graph nodes according to the relationship of the dependency syntax, and the frequency of the relationship as the edge weight. This means that our graph nodes are just words we extract from each sentence and some co-occurrences are used as edge weights. This has nothing to do with the input length of the sentence.

Reviewer 2 Report
Ð re-trained language models, represented by the bidirectional encoder representations from transformers (BERT), have achieved good success in machine reading comprehension (MRC). However, such models cannot effectively integrate significant features such as syntax relations, semantic connections, and long-distance semantics between sentences. This circumstance imposes significant restrictions on the ability of models to deeply understand complex texts in natural language.
In order for the BERT model to understand long texts, the authors suggested using graph neural network to feature textual sentences and in-sentence entities. The authors developed multigranularity syntax guidance (MgSG) module that consists of a ”graph with dependence” module and a ”graph with entity” module, and also suggested two graph structure construction methods using dependencies and named entities.
The authors analyzed the role of the dependencies and named entities in reading comprehension tasks, and demonstrated through experiments that both word and sentence granularities affect model performance.
The developed method was used on the Stanford Question Answering Dataset and outperforms traditional models in terms of both exact match (EM) and F score (F1).
As a comment, we can note the insufficiently detailed study of the number of dependency layers influence on the performance of the model. Ð’In the first rows of Table 4, there is a nonmonotonic change in the value of EM, therefore to exclude the possibility of repeating the nonmonotonicity effect, it would be necessary to carry out experiments with the number of layers equal to at least 6.
In general, the article is of undoubted interest and is recommended for publication after an additional experiments with the number of layers equal to 6.
Author Response
Thank you for the detailed review. We have carefully and thoroughly proofread the manuscript to correct all the grammar and typos. We would like to take this opportunity to thank you for all your time involved and this great opportunity for us to improve the manuscript. We hope you will find this revised version satisfactory.
Reviewer#2, Concern # 1:In the first rows of Table 4, there is a nonmonotonic change in the value of EM, therefore, to exclude the possibility of repeating the nonmonotonicity effect, it would be necessary to carry out experiments with the number of layers equal to at least 6.
Author response: Thanks for your professional suggestion. As you said, the results of this experiment show a non-monotonic trend. We think this is the result we expect. The purpose of this experiment is to find the optimal number of layers. When the gradient of the experimental result trend is 0, it is the optimal value. Finally, we have updated the experimental results of the sixth layer in the latest manuscript, thank you for your comments.

Reviewer 3 Report
The paper presents an approach leveraging a pre-trained language model like BERT, and extending it with explicit information about syntax dependency and named entities.
Contrary to what the authors claim in the introduction, the experimental evaluation (Tables 1 and 2) indicates that the preexisting models are sufficiently capable, and the proposed approach has a negligible effect on the performance measures. What is more, this effect is sometimes negative (Table 2), strongly indicating that the results are, by and large, spurious. The ablation study presented in Section 5.2 is surprising to say the very least, as Table 3 indicates that, if anything, the GE module (the part responsible for incorporating named entities) decreases the performance!
In the text, there are numerous undefined concepts and variables making it rather hard to follow. In my opinion, the description in its current form is not sufficient to replicate the results. The attached file contains many detailed comments (as PDF annotations) on what should be improved, particularly regarding the definitions and structure of the text. However, given the very weak experimental results, I believe the paper should be rejected due to the insufficient contribution to knowledge.

Author Response
Thank you for the detailed review. We have carefully and thoroughly proofread the manuscript to correct all the grammar and typos. For some format, pictures, citations and other issues, we will mark the newly submitted documents, and will no longer make explanations in the area below to save your review time. We would like to take this opportunity to thank you for all your time involved and this great opportunity for us to improve the manuscript. We hope you will find this revised version satisfactory.
Reviewer#3, Concern # 1:Line 230. How is the sliding step used?
Author response: Thanks for your professional suggestion. We are very sorry that our paper did not give you a clear understanding of its meaning. First, we will extract the entities in the input data, and after extracting the named entities, construct the extracted named entities as an undirected graph. The vertices in this graph are all named entities in the text, and the edges in the graph are the associations between named entities. In this paper, we use the co-occurrence relationship as the association between named entities: if two named entities appear at the same time in the same text segment, add an edge to the vertex corresponding to the two named entities. The text segment here are our proposed sliding windows. The sliding steps is the distance that the window slides each time. The figure below is a sample.
Reviewer#3, Concern # 2:Line 234. What is the range of named entities.
Author response: Thanks for your professional suggestion. According to the characteristics of machine reading comprehension task, we have extended the scope of named entities to include numeric entities and temporal entities.
Reviewer#3, Concern # 3:Line 247. The description seems to omit a fully connected layer between normalization and softmax.
Author response: Thanks for your professional suggestion. Exactly, indeed it is. Because the fully connected layer is not the focus we want to introduce, and it is also well known, we do not introduce it but reflect it in Equation 13.
Reviewer#3, Concern # 4:Table 2. Why report these if they are not used then with MgSG?
Author response: Thanks for your professional suggestion. First of all, we apologize because the experimental data of ELECTRA-base+MgSG and ELECTRA-large+MgSG cannot be displayed in the table due to our mistakes. These two experimental data existed before the manuscript was submitted but were omitted due to personal negligence. We have added it in the latest manuscript.
Reviewer#3, Concern # 5:Table3. GE actually decreases performance. What dataset?
Author response: We're sorry that we misunderstood you with our table. The +GE module in this table is not an addition based on the BERT+GD module. Instead, they are added separately to prove that both modules play a role in improving the performance of the model. So, the +GE module is not a performance degradation, it proves the correctness of our method anyway. We have also revised the representation of the table in the latest manuscript. And all ablation experiments were performed using the SQuAD dataset
Reviewer#3, Concern # 6:Line 332. This is not the probability of predicting an answer, this is the confidence of the model that the answer is correct
Author response: Thank you so much for paying attention to such tiny details and proposing the professional suggestion. In order to find the most accurate answer, we usually pick 10 candidate answers that are most likely to be the correct answer and select the most likely one. This is the evaluation indicators what I call probability and you call confidence. Since my purpose is to let people know that the evaluation indicators are actually screened when choosing an answer, I have made my explanation of this word too simplified. Whatever it's called, we know its purpose is to filter out the best answers and the change of this value will directly affect the model's choice of answer. The larger the value, the more the model will consider the candidate answer as the final answer, which can also be said to improve the model's confidence in the candidate answer. So here we can fully prove the effectiveness of our method.
Reviewer#3, Concern # 7:This is not an analysis at all. This is an incomplete example, as the question is missing.
Author response: Thanks for your professional suggestion. We provide a new example with body, question and answer. In this example, red is the correct answer, blue is the wrong answer, and green is the key letter in the question. First use the keywords (also entities) in the question to locate the corresponding answer area. Since the correct answer is not simply in the position of the subject or object, but in the interjection, and there are many misleading options around, this leads to the failure of BERT to answer correctly. In the dependency relationship, Woodward Park, features, and Shinzen Japanese Gardens are related.
We also verify the effectiveness of the dependency syntax through experiments. We will first average the hidden state representations of the last layer of BERT and generate the corresponding heatmap. We find that BERT is significantly more interested in regular subject position, but this is not where the answer comes from. We then summed and averaged the output of our module with the representation of the hidden state. The darker the color, the higher the impact. Due to the addition of dependency syntax, the attention of the model significantly weakens the position of the beginning, and is biased towards the parenthesis in the sentence, which is the source of the correct answer.
|
Fresno has three large public parks, two in the city limits and one in county land to the southwest. Woodward Park, which features the Shinzen Japanese Gardens, numerous picnic areas and several miles of trails, is in North Fresno and is adjacent to the San Joaquin River Parkway. Roeding Park, near Downtown Fresno, is home to the Fresno Chaffee Zoo, and Rotary Storyland and Playland. Kearney Park is the largest of the Fresno region's park system and is home to historic Kearney Mansion and plays host to the annual Civil War Revisited, the largest reenactment of the Civil War in the west coast of the U.S. |
|
Which is one of the park features located in North Fresno? |
|
BERT: Woodward Park BERT+MgSG: Shinzen Japanese Gardens(√) |
BERT
BERT + GD module
Reviewer#3, Concern # 8:The whole purpose was to incorporate additional semantics into the model so that it can understand more complex dependencies, but it couldn't understand more difficult grammatical structures.
Author response: Thanks for your professional suggestion. As we know, there are already a lot of studies on the SQuAD dataset, and it is very difficult to improve it, even a 1% improvement is gratifying. Different from conventional methods, our method utilizes graph structure and dependencies as support and achieves significant improvement on SQuAD. CMRC is a Chinese dataset, and different languages bring different grammatical rules and semantics. Compared with Chinese grammar rules, English grammar rules are obviously clearer, and the semantic information contained in the dependencies of words is also clearer, which can greatly help the understanding of sentences. Therefore, the improvement of the model for English data is also greater. But the grammatical structure of Chinese is more random and irregular, which makes it difficult for us to use the information contained in Chinese grammar accurately and correctly. On the other hand, it just proves that our model can make good use of the hidden information contained in explicit grammatical structures.
Although the improvement of the large model is not as large as that of the small model, there is still a relatively obvious improvement. For example, the EM of bert-larege has increased by 0.5%. Compared with the large improvement of the small model, because the learning ability of the large model is strong enough, some grammatical relations have been learned, which leads to the fact that the additional feature information we have may have been learned, so compared with the small model, the improvement is improved. is not very obvious. Then the improvement for albert-base+MgSG is better than that of bert-large, and the amount of parameters is much smaller than that of BERT-large. This proves that the improvement of our module is more obvious for small models, which provides a new idea for model thinning. Compared with the hundreds of millions of parameters of the pre-trained language model, the parameter amount is only increased by about 3M. it is completely negligible.
|
Model |
parameter |
inference time |
EM(%) |
F1(%) |
|
BERT-base |
110M |
164s |
75.8 |
79.2 |
|
BERT-base+MgSG |
113M |
182s |
77.2 |
81.8 |
|
BERT-large |
340M |
481s |
80.4 |
83.3 |
|
BERT-large+MgSG |
343M |
537s |
80.9 |
84.1 |
|
ALBERT-base |
11M |
176s |
77.1 |
80.1 |
|
ALBERT-base+MgSG |
14M |
193s |
81.5 |
84.8 |

Reviewer 4 Report
The work is interesting and presents a different idea to take advantage of the long-term context in an MRC.
However, the work in the current state is lacking in some points:
1. Please add some reasoning to consider QA input text, could this idea be used for any long enough textual format?
2. It is unclear how the graph models are trained, one of them appears to use Glove embeddings when BERT provides the main embeddings.
3. The Results only show a minuscule improvement over the traditional models. Please consider adding more datasets and statistical analyses of the results.
4. Most images have poor quality, consider using only scalable formats.
Author Response
Thank you for the detailed review. We have carefully and thoroughly proofread the manuscript to correct all the grammar and typos. For some format, pictures, citations and other issues, we will mark the newly submitted documents, and will no longer make explanations in the area below to save your review time. We would like to take this opportunity to thank you for all your time involved and this great opportunity for us to improve the manuscript. We hope you will find this revised version satisfactory.
Reviewer#4, Concern # 1: Please add some reasoning to consider QA input text, could this idea be used for any long enough textual format?
Author response: Thanks for your professional suggestion. As you know from the article, our proposed approach to accomplishing machine reading comprehension requires a pre-trained language model (such as BERT) and a graph-structured multigranularity syntax guidance module (MgSG). In theory, our MgSG module can accept text input of any length. Because 1. This module only needs to extract the syntactic dependency structure and named entities in the input text as the nodes of the graph structure. At the same time, we will use the words in it as the graph nodes according to the relationship of the dependency syntax, and the frequency of the relationship as the edge weight. This means that our graph nodes are just words we extract from each sentence and some co-occurrences are used as edge weights. This has nothing to do with the input length of the sentence. 2. However, due to the design limitations of the BERT model, the general pre-trained language model can only accept input with a length of 512 (of course you can choose to truncate long text, but this will cause irreversible huge semantic loss, we do not recommend doing so). At the same time, HPSG Neural Parser also has certain restrictions on the input length. This leads to the fact that as a whole model, it cannot accept inputs of arbitrary length. But in theory our MgSG can accept this kind of input.
Reviewer#4, Concern # 2: It is unclear how the graph models are trained, one of them appears to use Glove embeddings when BERT provides the main embeddings.
Author response: Thanks for your professional suggestion. We are very sorry that you cannot understand the usage of this part because we do not explain in detail. Our input text is to be fed into two parts. Part of them is to be sent to the graph structure (as shown in Figure 1). GloVe is used here as a word vector. When our selected word is used as a graph node, a digital matrix is required as the expression form of the word. Due to the advantage of GloVe being more global in training, we use it as a word vector and update the node. We made changes around line 203 in the latest submitted manuscript.
“GloVe[ 8] is used as the embedding representation for each token as a feature of the nodes; sentence-based dependencies are used as association relations. If the current token is not in GloVe’s vocabulary, we will skip the current token and not act as a node.”
Reviewer#4, Concern # 3:The Results only show a minuscule improvement over the traditional models. Please consider adding more datasets and statistical analyses of the results.
Author response: Thanks for your professional suggestion. As the most famous public dataset, SQuAD is very interesting and difficult for enterprises and professionals all over the world. As far as we know, for this task, an improvement of even 1% is exciting enough to innovate and modify pre-trained language models. However, we chose to use the characteristics of graph structure to creatively use both named entities and syntactic dependencies as features in the MRC task and achieved a good improvement. It provides new features and new ideas for people to study this field later. In addition, we would like to give a more in-depth explanation on SQuAD of the effectiveness of the method proposed in this paper in terms of parameter quantity and model efficiency.
From Table 1, we can see that the parameter amount of the method proposed in this paper is only related to the number of graph nodes (that is, the number of selected feature words), and the parameter amount is only increased by about 3M. Compared with the hundreds of millions of parameters of the pre-trained language model, it is completely negligible. Due to the negligible number of parameters, we found that the increase in inference time of the model was also minimal. Compared with the increase in parameters and inference time of BERT-large, we can say that there is almost no increase in computational cost.
Finally, we attach a related paper of AAAI in 2020, the specific content is not introduced. The paper also uses the SQuAD dataset, and the improvement of SG-Net compared to the baseline is 1%, which is an exciting improvement. The improvement of our corresponding model is 1.4%, so the improvement in this paper is not insignificant.
|
Model |
parameter |
inference time |
EM(%) |
F1(%) |
|
BERT-base |
110M |
164s |
75.8 |
79.2 |
|
BERT-base+MgSG |
113M |
182s |
77.2 |
81.8 |
|
BERT-large |
340M |
481s |
80.4 |
83.3 |
|
BERT-large+MgSG |
343M |
537s |
80.9 |
84.1 |
|
ALBERT-base |
11M |
176s |
77.1 |
80.1 |
|
ALBERT-base+MgSG |
14M |
193s |
81.5 |
84.8 |
Zhang Z, Wu Y, Zhou J, et al. SG-Net: Syntax-guided machine reading comprehension[C]//Proceedings of the AAAI Conference on Artificial Intelligence. 2020, 34(05): 9636-9643.
Reviewer#4, Concern # 4: Most images have poor quality, consider using only scalable formats.
Author response: Thanks for your professional suggestion. We're sorry to have you comment on the unclear image, we've redrawn the image more to ensure its clarity.

Round 2
Reviewer 1 Report
Thank you kindly for you thorough revision and detailed responses.
Author Response
Dear reviewer,
Thank you very much for your reply and recognition of our work, I wish you all the best in your life and work.

Reviewer 3 Report
The authors addressed my minor concerns. The major concern that the proposed method offers no real improvement remains - the results reported in Tables 1 and 2 indicate that for some models there is a decrease in the performance, indicating in turn that the proposed approach offers no real benefit, but is a random change for the models. In this aspect, my former recommendation of rejecting the paper stands. As this seems to be a non-issue for the editor, I think the paper is suitable for publication in the present form.
Author Response
Reviewer#3, Concern # 1:the results reported in Tables 1 and 2 indicate that for some models there is a decrease in the performance, indicating in turn that the proposed approach offers no real benefit, but is a random change for the models.
Author response: Thank you for your reply. As you have noticed, the experimental results of the paper have little improvement or even a decrease in performance on Chinese data and some large models. We also noticed this phenomenon during the experiment and explained it accordingly in the manuscript. One of the innovations of our manuscript is to extract useful information for understanding from grammatical relations and improve model performance. Those languages with strict grammar and explicit syntactic relations can better benefit from the proposed approach, for example, on the ALBERT-base Model, the EM is improved from 77.1% to 81.5%. The reason for the insignificant improvement or even declines in Chinese is that Chinese is a language with more idioms than grammatical constraints, which means that Chinese does not strictly follow grammatical rules. Therefore, the model can extract less information from the grammatical relationship that is helpful for the model to understand, so the improvement effect is not as good as English. However, this also proves that our method can well use the hide information contained in the clear grammatical structure. The improvement of unclear grammatical structure will be the focus of our follow-up research.
Thanks for all your professional advice.
Best wishes to you.

Reviewer 4 Report
From my previous concerns, only part of them was addressed: 4) and 1) (I would like the response given to be added to the document).
The other concerns were not adequately addressed:
2) The graph training method is still not straightforward, do you have a threshold for selecting an edge between two words? The embeddings from GLoVe can be used to estimate similarity. Why do you state that GloVe embeddings are more global? It is widely accepted that bert embeddings provide a better translation into a feature space.
3) My comment regarding the improvements still stands. 1% is a relatively small improvement. You presented another article with a 1.4% improvement. That has two issues, their improvement is better than the one in this paper. Second, that conference does not appear to be indexed by any major platform.
Regardless, my comment can be simply resolved by adding a second dataset or performing a simple statistical validation that shows that the 1% is not a random event but a proper improvement.
Given the presented paper, we would like to see and improvement larger than 1.4%
Author Response
Thank you for the detailed review. We have carefully and thoroughly proofread the manuscript to correct all the grammar and typos. For some format, pictures, citations and other issues, we will mark the newly submitted documents, and will no longer make explanations in the area below to save your review time. We would like to take this opportunity to thank you for all your time involved and this great opportunity for us to improve the manuscript. We hope you will find this revised version satisfactory.
Reviewer#4, Concern # 1:The graph training method is still not straightforward; do you have a threshold for selecting an edge between two words?
Author response: Thanks for your professional suggestion. At the same time, we are very sorry that we did not solve your doubts in round1's reply. First, in the graph with dependency module, our input will generate multiple sets of dependencies after dependency analysis, and each set has multiple words with hierarchical relationships(as shown in the figure below,only part of the sets are listed). Here we will select these words as graph nodes, and the weight of the edge is the number of co-occurrences of the two words, that is to say, when the two words appear in different sentences, the edge weight will add 1. Notably, we set a threshold here to constrain our choice of words. The threshold here is used to limit the hierarchical relationship of words in the above-mentioned set. Through our experiments, it is proved that setting a certain threshold will improve the performance of the model (the experimental results are shown in Table 5).
Then in the graph with entity module, we extract the entities in the sentence. The word is also used as the graph node of the graph neural network, and then the word vector fused in the Integration module is represented as the graph node vector. Then take the co-occurrence times of words and words in the sliding window as the edge weight. The specific process is shown in the figure below. That's the training details of our graph, hopefully that explains it for you. We also supplement the description of graph structure training in our latest manuscript
Reviewer#4, Concern # 2:The embeddings from GLoVe can be used to estimate similarity. Why do you state that GloVe embeddings are more global? It is widely accepted that bert embeddings provide a better translation into a feature space.
Author response: Thanks for your professional suggestion. Sorry for the unclear answer in my reply in round1. The GloVe mentioned here is more global than word2vec. In Word2Vec, the model is designed based on the assumption that the surrounding words are more semantically related to the central word, but the word frequency statistics are not considered. GloVe uses the characteristics of co-occurrence matrix and word vector matrix to obtain word vector matrix with stronger semantic similarity after multiple rounds of training. Related content is also mentioned in the original paper. The reason why we did not use bert here is mainly to consider the computational cost of the model and the lightweight of the model. The stacking of multiple pre-trained language models is not the original intention of our research. At the same time, GloVe is used as the representation of word vectors in many studies, such as TinyBERT.
Pennington J, Socher R, Manning C D. Glove: Global vectors for word representation[C]//Proceedings of the 2014 conference on empirical methods in natural language processing (EMNLP). 2014: 1532-1543.
Xiaoqi Jiao, Yichun Yin, Lifeng Shang, Xin Jiang, Xiao Chen, Linlin Li, Fang Wang, and Qun Liu. 2020. TinyBERT: Distilling BERT for Natural Language Understanding. In Findings of the Association for Computational Linguistics: EMNLP 2020, pages 4163–4174, Online. Association for Computational Linguistics.
Reviewer#4, Concern # 3:my comment can be simply resolved by adding a second dataset or performing a simple statistical validation that shows that the 1% is not a random event but a proper improvement.
Author response: Thanks for your professional suggestion. First of all, I would like to correct some of your doubts. First, SG-Net has been published in the AAAI-2020 conference and is also available in the conference proceedings and Google Scholar. As of today, the paper has been cited 122 times. In addition, compared to the 1% improvement of SG-net in the paper, the improvement of our method is 1.4%, which is higher than that of SG-net.
It is difficult to find authoritative datasets and perform data preprocessing, analysis and extraction of dependencies and other related work in a short period of time. In addition, the extensive experiments are required for hyperparameter selection which is time-consuming. At the same time, the experimental results in our manuscript are already the results of 2-3 experiments. In our experimental results, we found that the performance of the model is stable, so we did not specifically indicate it in the manuscript. To dispel your doubts about the randomness of model performance, we decided to run 5 experiments on some of the models in the SQuAD dataset, without setting a random seed. The experimental results are shown in the figure below, and the experimental data are the mean and standard deviation.
Zhang, Z., Wu, Y., Zhou, J., Duan, S., Zhao, H. and Wang, R. 2020. SG-Net: Syntax-Guided Machine Reading Comprehension. Proceedings of the AAAI Conference on Artificial Intelligence. 34, 05 (Apr. 2020), 9636-9643. DOI: https://doi.org/10.1609/aaai.v34i05.6511.
Conference URL: https://ojs.aaai.org/index.php/AAAI/article/view/6511
|
Model |
EM(%) |
F1(%) |
EM(%) for MgSG |
F1(%) for MgSG |
|
BERT-base |
75.8 |
79.2 |
77.32 (±0.12) |
81.13 (±0.09) |
|
BERT-large |
80.4 |
83.3 |
81.04 (±0.10) |
84.04 (±0.08) |
|
ELECTRA-base |
79.5 |
82.5 |
80.32 (±0.10) |
83.67 (±0.10) |
|
ALBERT-base |
77.1 |
80.1 |
81.57 (±0.13) |
84.83 (±0.11) |
|
ALBERT-large |
79.4 |
82.3 |
81.86 (±0.11) |
85.60 (±0.10) |
|
XLNET-base |
77.6 |
80.3 |
79.63 (±0.13) |
82.23 (±0.09) |
